# Novel Applications of Lactic Acid and Acetic Acid Bacteria Preparations in Shaping the Technological and Microbiological Quality of Ready-to-Cook Minced Pork

**DOI:** 10.3390/foods14111934

**Published:** 2025-05-29

**Authors:** Marcelina Karbowiak, Anna Okoń, Beata Łaszkiewicz, Piotr Szymański, Dorota Zielińska

**Affiliations:** 1Institute of Human Nutrition Sciences, Warsaw University of Life Sciences (SGGW), Nowoursynowska 159c, 02-776 Warsaw, Poland; 2Department of Meat and Fat Technology, Institute of Agricultural and Food Biotechnology—State Research Institute (IBPRS-PIB), Rakowiecka 36, 02-532 Warsaw, Poland; anna.okon@ibprs.pl (A.O.); beata.laszkiewicz@ibprs.pl (B.Ł.); piotr.szymanski@ibprs.pl (P.S.)

**Keywords:** biopreservation, lactic acid bacteria (LAB), acetic acid bacteria (AAB), meat shelf life, ready to cook, ground meat, pork, natural preservatives, food safety

## Abstract

This study explored a novel application of bacterial preparations, derived from lactic acid bacteria (LAB) and acetic acid (AAB), to preserve ready-to-cook minced pork. Two LAB and AAB cell-free supernatant mixtures were evaluated as raw meat additives during nine refrigerated storage days. Both treatments effectively stabilized the meat’s pH (final values around 5.54) and oxidation reduction potential (final ORP values around 336–349 mV), while preserving color parameters (*L**, *a**, *b**) without significant degradation. Lipid oxidation, measured by TBARS, was significantly reduced in treated samples (0.34–0.37 mg MDA/kg) compared to the control (0.43 mg MDA/kg) by day 9. Microbial counts were markedly lower: total viable counts in treated samples did not exceed 3.2 log CFU/g, whereas the count in the control reached 4.6 log CFU/g. Exploratory factor analysis (EFA) revealed that microbial growth was the dominant factor affecting quality deterioration, while lipid oxidation and color stability formed distinct quality axes. Functional principal component analysis (FPCA) showed that among treatments, the combination of *Lactiplantibacillus plantarum* O24 and *Gluconobacter oxydans* KNS32 (T2) demonstrated the most effective biopreservation, achieving the best microbiological and oxidative stability. This study introduces the novel, synergistic use of LAB and AAB preparations as a clean-label biopreservation strategy for addressing minced meat products.

## 1. Introduction

Meat, particularly in its raw and minimally processed forms, is among the most perishable food products due to its high moisture content, rich nutritional profile, and susceptibility to microbial contamination and oxidative degradation [1]. Despite advances in cold chain logistics and packaging, maintaining the quality and safety of meat throughout storage remains a significant challenge for the food industry. Spoilage-related changes compromise sensory and nutritional quality and reduce shelf life, posing economic and public health concerns [2]. In recent years, ready-to-cook meat products—such as pre-packaged minced meat—have gained popularity due to shifting consumer lifestyles, increased demand for convenience, and changing cooking habits [3]. Ready-to-cook minced meat offers the advantage of minimal preparation, but its high surface area and lack of thermal processing increase vulnerability to microbial proliferation and lipid oxidation [4]. As a result, effective preservation strategies are crucial to ensure product safety and technological quality [5].

At the same time, consumer expectations are evolving, with a growing preference for “clean-label” products that contain fewer synthetic additives and more natural, recognizable ingredients [6]. This shift has intensified interest in alternative preservation strategies, particularly those based on natural antimicrobials such as plant extracts, essential oils, and microbial metabolites [6]. One promising response to these demands is the concept of biopreservation, which involves using natural or controlled microbiota—or their antimicrobial metabolites—to extend the shelf life and enhance the safety of foods [7]. Originating from traditional fermentation processes, biopreservation relies on inhibitory compounds such as organic acids, bacteriocins, and hydrogen peroxide, produced by beneficial microorganisms, mainly lactic acid bacteria (e.g., *Lactococcus*, *Lactobacillus*, *Pediococcus*, *Leuconostoc*, *Carnobacterium*, *Enterococcus*, and *Streptococcus*) and *Bifidobacterium* [2]. Unlike chemical preservatives, biopreservatives are typically recognized as safe and can offer dual benefits: protection against spoilage and the maintenance of nutritional or sensory quality [8,9]. The use of biopreservation has expanded in recent decades beyond fermented foods. It is now increasingly applied in minimally processed products, including fresh meats, seafood, and dairy, where it can serve as a clean-label alternative to synthetic additives [8,9]. Moreover, this approach supports the development of functional foods, products that offer health benefits beyond basic nutrition through bioactive compounds [3]. While the current study does not directly evaluate health benefits, incorporating bacterial metabolites, especially those associated with probiotic activity in traditional fermented foods like dairy products or kombucha, reflects the functional food philosophy and consumer interest in products with added biological value [10].

Among microbial sources, lactic acid bacteria (LAB) and their metabolites have received considerable attention due to their ability to inhibit spoilage and pathogenic microorganisms [11]. Acetic acid bacteria (AAB) have also emerged as a microbial group with promising technological potential, being traditionally recognized for their role in vinegar production and kombucha fermentation [12,13]. Similarly to LAB, AAB produce a diverse range of organic acids and redox-active compounds with documented antimicrobial properties [14]. Despite this, their use in meat systems remains limited, particularly in the form of cell-free supernatants or metabolic extracts [15]. In response to these trends, the combined application of metabolite-rich supernatants from both LAB and AAB strains represents an innovative biopreservation strategy, aligned with the demand for natural and effective methods in meat processing. While LAB-derived supernatants have been previously studied in meat products [16,17,18,19], the originality lies in the synergistic use of LAB and AAB metabolites, a concept inspired by the microbial interactions observed in kombucha, where these bacteria coexist in a symbiotic culture and jointly produce a spectrum of bioactive compounds that stabilize microbial ecosystems and enhance sensory properties [20]. However, despite growing interest in biopreservation, research exploring the combined application of LAB and AAB metabolites in meat systems remains scarce, particularly in the context of ready-to-cook products. It should also be noted that these LAB+AAB combinations have already been tested in vitro in preliminary studies (not yet published), indicating promising antimicrobial efficacy and potential for application in meat preservation. Although the antimicrobial properties of LAB-derived supernatants have been previously explored, the synergistic application of LAB and AAB metabolites for the preservation of minimally processed ready-to-cook meat remains largely under-investigated. This study pioneers the combined use of LAB and AAB cell-free supernatants, offering a novel, clean-label solution to enhance microbiological safety and oxidative stability in minced meat products.

In line with current trends toward natural meat preservation, this study aimed to develop a pioneering approach, utilizing food-grade bacterial preparations rich in antimicrobial metabolites, to enhance the safety and quality of ready-to-cook minced pork. Specifically, the research evaluated the effects of synergistic LAB and AAB cell-free supernatant mixtures on key technological and microbiological parameters during refrigerated storage. This study addresses a critical gap concerning the application of LAB+AAB biopreservatives in minimally processed meat systems and proposes an innovative natural preservation strategy aligned with clean-label and functional food market demands.

## 2. Materials and Methods

### 2.1. Bacterial Strains, Culture Media, and Growth Conditions

Four bacterial strains were selected from the internal microorganism collection of the Institute of Human Nutrition Sciences, Warsaw University of Life Sciences (SGGW), based on prior evidence of antimicrobial and functional activity. Two lactic acid bacteria (LAB) strains, *Lacticaseibacillus paracasei* B1 and *Lactiplantibacillus plantarum* O24, were initially isolated from Polish regional sheep’s milk cheese (Bundz, GenBank accession no: CP161807, CP161808) and traditionally fermented cucumber dill pickles (GenBank accession no: CP157747-CP157755), respectively. Two acetic acid bacteria (AAB) strains, *Gluconobacter oxydans* KNS32 (GenBank accession no OQ597203) and *Komagataeibacter saccharivorans* KOM1 (GenBank accession no OQ594827), were derived from a local kombucha beverage and artisanal honey vinegar, respectively. The strains have been previously tested and have shown in vitro probiotic and anti-cancer properties [21,22,23,24].

All strains were cryopreserved at −80 °C in their respective growth media, supplemented with 20% (*v*/*v*) glycerol. LAB strains were maintained in de Man, Rogosa, and Sharpe (MRS) broth (Neogen, Lansing, MI, USA). In contrast, AAB strains were preserved in Glucose Calcium Carbonate (GC) broth composed of 2% *w*/*v* glucose (Merck Life Science, Darmstadt, Germany), 0.7% *w*/*v* calcium carbonate (Merck Life Science, Darmstadt, Germany), 0.3% *w*/*v* yeast extract (Merck Life Science, Darmstadt, Germany), and 0.3% *w*/*v* casein peptone (Merck Life Science, Darmstadt, Germany).

For cultivation, LAB strains were grown anaerobically in freshly prepared MRS broth at 37 °C for 16–18 h, reaching a final concentration of approximately 1 × 10^9^ CFU/mL. Anaerobic conditions were maintained using AnaeroGen™ sachets (Oxoid, Basingstoke, UK). Following incubation, LAB cultures were heat-treated in a water bath at 80 °C for 20 min. AAB strains were cultivated aerobically in a Hestrin and Shramm (HS) medium at 28 °C for 48 h, achieving a bacterial density of 1–5 × 10^7^ CFU/mL. The HS medium contained 2% (*w*/*v*) glucose (Merck Life Science, Darmstadt, Germany), 0.5% (*w*/*v*) casein peptone (Merck Life Science, Darmstadt, Germany), 0.5% (*w*/*v*) yeast extract (Merck Life Science, Darmstadt, Germany), 0.27% (*w*/*v*) Na_2_HPO_4_ (Pol-Aura, Morąg, Poland), and 0.15% (*w*/*v*) citric acid (Merck Life Science, Darmstadt, Germany). Bacterial suspensions were centrifuged at 3341× *g* for 15 min (Eppendorf SE, Hamburg, Germany) to separate bacterial cells. The supernatants were subsequently sterilized using 0.22 μm syringe filters (AlfaChem, Poznań, Poland) and stored at −20 °C for up to 14 days before further analysis. To address potential variability, bacterial cultures were standardized by employing defined media, controlled incubation parameters, and consistent cell density. The same preparation procedure was applied across all replicates. Additionally, heat-treated, cell-free supernatants derived from defined strains were used to ensure batch-to-batch consistency and to eliminate the variability associated with ongoing microbial metabolism during storage. Finally, two types of preparations, LAB cell-free supernatants of heat-killed cells (hkCFS) and AAB cell-free supernatants (CFS), were assembled. The supernatants and their combination were selected based on previous studies demonstrating their synergistic action in vitro and in situ. The physicochemical characteristics of these preparations were analyzed. The pH values of the LAB and AAB preparations ranged from 3.2 to 4.5, with the lowest values observed in LAB-derived cell-free supernatants. Among organic acids, lactic acid (approx. 8.3 mg/mL) and acetic acid (approx. 6.7 mg/mL) were predominant. Gluconic acid was particularly abundant in AAB preparations (approx. 5.9 mg/mL).

### 2.2. Meat Sample Preparation and Experimental Design

The raw material used for the study consisted of lean meat and fat from pork ham, minced using an 8 mm mesh. The meat was excised 48 h post-mortem from carcasses chilled at 2 °C. The raw material used had no quality defects. The chemical composition of the minced meat was as follows (mean values ± SD for three batches of raw material): water content [%]—60.95 ± 1.83, total protein content [%]—16.00 ± 0.90, collagen content [%]—1.36 ± 0.22, and fat content [%]—22.48 ± 2.25. The meat was obtained from a local, medium-sized processing plant near Warsaw, Poland. Experimental production was conducted in the technical hall of the Department of Meat and Fat Technology at the Institute of Agricultural and Food Biotechnology in Warsaw (Poland). The meat was subdivided into uniform portions for analysis. Raw minced pork meat samples were randomly assigned to three experimental groups: two treatment groups (T1 and T2) and one control group (C). Treatment group T1 received a formulation containing 1.25% (*w*/*w*) heat-killed cell-free supernatant (hkCFS) of *Lacticaseibacillus paracasei* B1, 1.25% (*w*/*w*) cell-free supernatant (CFS) of *Komagataeibacter saccharivorans* KOM1, 2.5% (*v*/*v*) distilled water, and 1% (*w*/*w*) salt. Treatment group T2 was supplemented with 1.25% (*w*/*w*) hkCFS of *Lactiplantibacillus plantarum* O24, 1.25% (*w*/*w*) CFS of *Gluconobacter oxydans* KNS32, 2.5% (*v*/*v*) distilled water, and 1% (*w*/*w*) salt. The control group (C) consisted of samples without any bacteria-derived supplementation and was formulated with 5% (*v*/*v*) distilled water and 1% (*w*/*w*) salt (Figure 1). The concentration of cell-free supernatants (CFS) in treatments T1 and T2 was set at 2.5% (*w*/*w*), supplemented with 2.5% (*v*/*w*) distilled water to maintain a total additive level of 5%. This proportion reflects standard practice when preparing ready-to-cook minced meat products, where up to 5% water is typically added to improve texture and processing characteristics [25]. Preliminary in-house trials indicated that higher CFS concentrations adversely affected product quality, while 2.5% provided a functional effect without compromising sensory attributes. Similar concentrations were reported as effective in other studies using bacteria-derived supernatants in meat matrices [19]. For each treatment condition, the respective components were thoroughly mixed with a batch of raw pork meat to obtain a homogeneous mixture, which was then aseptically transferred into sterile polyethylene bags and vacuum-sealed into plastic casings to produce three (n = 3) individual replicates per group. This procedure used three pork meat batches (lots) on separate days. Samples were stored at 4 ± 1 °C in a controlled cold room environment and analyzed at 0, 3, 6, and 9 days post-processing to evaluate changes over time. Time and storage conditions were established based on preliminary research. Time-zero samples were analyzed approximately two hours after processing. Figure 1 shows an overview of the experimental design and model appearance of the study’s minced pork samples after preparation.

### 2.3. Physicochemical Analyses

#### 2.3.1. The pH Measurement

The pH measurement was performed according to the methodology described by Łaszkiewicz et al. (2021) [26]. The 10 g meat product samples were mixed with 50 mL distilled water and homogenized (14.000 RPM) for 1 min with an 800 W blender (MSM 66120, BSH Hausgeräte GmbH, Munich, Germany). To determine the pH value, we used a digital pH meter (Mettler Delta 350, Mettler Toledo, Schwerzenbach, Switzerland) with the automatic compensation of temperature and a glass calomel in a Lab Cool electrode (Mettler Toledo, Greifensee, Switzerland).

#### 2.3.2. Oxidation Reduction Potential (ORP) Measurement

To determine the ORP value, the 10 g meat product samples were mixed with 50 mL distilled water and homogenized (14.000 RPM) for 1 min with an 800 W blender (MSM 66120, BSH Hausgeräte GmbH, Munich, Germany). The redox potential was assessed using a digital pH meter (Mettler Delta 350, Mettler Toledo, Schwerzenbach, Switzerland) equipped with an In Lab Redox Pro electrode (Mettler Toledo, Greifensee, Switzerland). The results obtained were transformed into the value of ORP in relation to the standard hydrogen electrode EH (mv). The ORP value of the reference electrode at a temperature of 20 °C—Eref = 207 mV was summed with the value obtained with the equipment.

#### 2.3.3. Color Measurement

A spherical CR-300 spectrophotometer (Konica Minolta, Tokyo, Japan), with a measuring hole 25.4 mm in diameter, was used to determine the color. The measurement was performed in a laboratory at 24 °C ± 2 °C. Computed trio-chromatic coordinates were expressed in the CIE *L** *a** *b** system, where *L** means lightness, *a** means chromaticity from green to red, and *b** means chromaticity from blue to yellow. For the measurement of the color, the standard CIE observer was used. This has the following parameters: 2°, illuminant D65, and 8 mm measuring area. The white tiling standard (*L** = 95.87, *a** = −0.49, *b** = 2.39) was used as a reference source. Six measurements were carried out for each of the three replicate treatments.

### 2.4. Microbiological Analysis

Bacterial enumeration and pathogen detection were performed at 0, 3, 6, and 9 days of refrigerated storage. For each sample, 10 g or 25 g of meat was aseptically transferred into a sterile Stomacher bag and diluted with 90 mL or 225 mL of sterile buffered peptone water (Oxoid, Basingstoke, UK) to obtain an initial 10-fold dilution. Samples were blended at 230 rpm using a Stomacher 400 instrument (IUL Instruments, Königswinter, Germany) for 2 min, followed by serial decimal dilutions and plating on selective media. The total viable count (TVC) was determined using Plate Count Agar (PCA) (Merck Life Science, Darmstadt, Germany), incubated at 30 °C for 48 h. Lactic acid bacteria (LAB) were enumerated on MRS agar (Neogen, Lansing, MI, USA) and incubated at 37 °C for 48 h. *Escherichia coli* was detected using Tryptone Bile X-Glucuronide (TBX) agar (Oxoid, Basingstoke, UK), incubated at 44 °C for 18–24 h. Enterobacteriaceae were plated on Violet Red Bile Glucose (VRBG) agar (Oxoid, Basingstoke, UK) and incubated at 37 °C for 24 h. Coagulase-positive *Staphylococcus* were detected using a Rabbit Plasma Fibrinogen (RPF) (Biokar Diagnostics, Allonne, France) medium incubated at 37 °C for 18–24 h. *Salmonella* spp. detection involved enrichment in Müller–Kauffman’s tetrathionate broth with novobiocin (Oxoid, Basingstoke, UK), followed by plating on XLD agar (Oxoid, Basingstoke, UK) and Rambach chromogenic medium (Merck KGaA, Darmstadt, Germany), incubated at 37 °C for 24 h. *Listeria monocytogenes* was enriched in Half Fraser and Fraser broths with the X211 supplement (Neogen, Lansing, MI, USA) followed by plating on Listeria chromogenic agar with ×10 and ×72 supplements (Neogen, Lansing, MI, USA) and Palcam agar (Neogen, Lansing, MI, USA), incubated at 37 °C for 24–48 h. Bacterial counts were calculated and expressed as log_10_ CFU/g. In cases where no colonies were observed on the plate corresponding to the lowest dilution, a value of one colony was assumed for calculation purposes.

### 2.5. Lipid Oxidation and Fatty Acid Analysis

Lipid oxidation was assessed by determining thiobarbituric acid reactive substances (TBARS), following the methods of Pikul et al. (1989) [27] and Szymański et al. (2025) [28]. For each sample, 10 g of meat was homogenized with 34.25 mL of 4% cold perchloric acid and 750 μL of a 0.01% ethanolic BHT solution using a Bamix m200 blender (ESGE AG, Mettlen, Switzerland) for 1 min. The resulting homogenate was filtered, and 1 mL of the filtrate was combined with 1 mL of 0.02 M aqueous 2-thiobarbituric acid. The mixture was then incubated at 100 °C for 60 min. After cooling to approximately 20 °C, absorbance was measured at 532 nm using a U-2900 spectrophotometer (Hitachi, Tokyo, Japan). A blank sample of 1 mL of 4% perchloric acid and 1 mL of 0.02 M thiobarbituric acid solution served as the reference. TBARS values were expressed as milligrams of malondialdehyde (MDA) per kilogram of sample.

Cholesterol content was determined using a gas chromatograph equipped with a flame ionization detector (HP/Agilent 6890 II-FID) (Agilent Technologies, Santa Clara, CA, USA), following the IBPRS-PIB laboratory’s internal procedure (PA/04, Issue 7, dated 8 June 2021) [29], with results expressed in mg per 100 g of product.

The fatty acid composition was determined by gas chromatography using ISO 12966-1:2014 [30]. The analytical procedures followed methods similar to those described by [28]. Gas chromatographic analysis was performed using a flame ionization detection system (HP 6890 II-FID; Agilent Technologies, Santa Clara, CA, USA) equipped with a BPX 70 column (Trajan Scientific and Medical, Ringwood, VIC, Australia) featuring a highly polar stationary phase. The column specifications were as follows: a length of 100 m, a film thickness of 0.20 μm, and an internal diameter of 0.25 mm. Heneicosanoic acid (C21:0; H5149, Sigma-Aldrich, St. Louis, MO, USA) was used as an internal standard. Individual fatty acids were identified by comparing retention times with the certified reference material Supelco 37-Component FAME Mix (CRM47885, Merck KGaA, Darmstadt, Germany). The resulting data are given in g/100 g total fatty acid.

### 2.6. Statistical Analysis

This study tested parameters across four time points: baseline (day 0), day 3, day 6, and day 9. Performance results were reported as medians with interquartile ranges. The Friedman test was applied to assess the effect of storage duration, while the Kruskal–Wallis test was used to evaluate treatment effects, both at a significance level of α = 0.05. Nonparametric tests were chosen due to the violation of the normality assumption in the dataset. Pairwise median comparisons were performed using the Durbin–Conover test and the Dunn–Sidak correction factor (DSCF). For multivariate exploratory analysis, exploratory factor analysis (EFA) was employed. Due to deviations from the normal distribution, the principal axis factoring method was applied, with Oblimin rotation used to account for correlations among factors. We determined the number of factors using the eigenvalue > 1 criterion, and factor loadings above 0.50 were considered significant. Analyses were performed for days 0 and 9, as complete data were available for these storage time points. To observe changes in the values of the studied parameters over time, functional principal component analysis (FPCA) was conducted. FPCA is a statistical method used to analyze data that changes continuously (e.g., over time). The aim of FPCA was to identify the main patterns of variability within functions (continuous variables). These patterns, called functional principal components, allow for a simplified representation and understanding of the changes occurring in the data. Suppose that the secured stochastic process X(t) is subject to the period [0,T] and that, for this process, I use the covariance c(s,t) for s,t∈T. Theoretically, the process $X\left(t\right)\;$ can be applied as follows:(1)$$X\left(t\right)=\sum_{j=1}^{\infty }\varepsilon_{j}c_{j}(t),$$
where $c_{j}(t)$ is a basis in the space of square-integrable functions defined on the interval [0,T] $L^{2}[0,T]$, and $\varepsilon_{j}$ are random variables [31].

The main goal is to choose the appropriate basis $b_{j}(t)$ and random variables $\eta_{j}$ in the following way:(2)$$\int_{0}^{T}b_{j}^{2}\left(t\right)dt=1,\;\int_{0}^{T}b_{j}\left(t\right)b_{k}\left(t\right)dt=0\;(j\neq k)$$
The variances were non-decreasing $\left(D_{\eta_{1}}^{2}>D_{\eta_{2}}^{2}>\ldots \right)$. The elements of this basis are called functional principal components [31].

Since the parameters describing the individual properties of the meat samples were expressed in different units, data standardization was performed. Furthermore, missing data were supplemented using linear interpolation.

Statistical calculations were performed using Jamovi (The Jamovi Project, version 2.6.26) and RStudio 2024.12.1-563 (Posit, Boston, MA, USA).

## 3. Results and Discussion

### 3.1. Determination of pH and Oxidation Reduction Potential (ORP) Measurement

Table 1 presents the pH and ORP values of minced pork samples during storage.

At the beginning of storage (day 0), the pH values of the treated samples (T1 and T2) were comparable to those of the control group (*p* > 0.05), indicating that the addition of LAB+AAB supernatants did not cause any immediate or significant acidification. Despite the presence of antimicrobial organic acids such as lactic, acetic, and gluconic acids in the supernatants [21,32], treated samples exhibited a moderate, gradual pH decline up to day 6, reaching minimum values around 5.38. After day 6, a slight pH increase was observed in all samples. This was probably associated with proteolytic and enzymatic activities linked to ongoing microbial metabolism [33]. Notably, the dynamics of pH changes remained closely aligned with those observed in the control group, indicating that the antibacterial supernatants exerted their preservative action without substantially altering the physicochemical balance of the meat. This observation is important from a technological standpoint, as it preserves the isoelectric point of meat proteins, maintaining water-holding capacity and preventing excessive fluid losses during thermal processing [34]. In contrast, conventional preservation methods based on the direct addition of organic acids often trigger a substantial pH decline, which, although effective against spoilage microorganisms, can impair protein functionality and result in higher economic losses due to reduced cooking yields [35,36]. This controlled acidification likely results from the absence of viable acid-producing bacteria and possible interactions between bioactive metabolites and the meat matrix, leading to a buffered effect [6]. Nevertheless, throughout the storage period, the treated samples maintained pH values comparable to the control, supporting the conclusion that LAB+AAB supernatants can extend shelf life without negatively impacting essential technological properties. These findings are consistent with previous research reporting that biopreservatives based on bacterial metabolites can effectively maintain microbial safety while preserving functional and sensory meat quality [17,19].

In all samples, a general increase in ORP was observed from day 0 to day 6, indicating progressive oxidation within the meat matrix. In the treated samples T1 and T2, which contained antibacterial supernatants, ORP values peaked on day 6 and significantly (*p* < 0.05) declined by day 9. This pattern may suggest some modulation of oxidative processes by the supernatant-derived compounds, possibly due to their antioxidant or microbial-inhibitory effects [37]. In contrast, the control sample exhibited a significant (*p* < 0.05) increase in ORP by day 9, reaching the highest recorded median (374.5 mV). This trend is consistent with typical oxidative processes during chilled storage, as oxygen availability and microbial metabolism can shift the redox balance toward more oxidizing conditions [38]. This sharp rise could reflect enhanced oxidative stress, likely driven by intensified microbial activity and associated metabolic byproducts such as peroxides or reactive oxygen species [39]. The absence of any antimicrobial intervention in the control sample may have allowed for the more rapid progression of spoilage and oxidation [40]. The ORP trends observed in this study align with previous findings linking microbial growth and oxidative deterioration in meat products during storage [41]. The less pronounced ORP changes in treated samples suggest the potential mitigating effect of the culture supernatants on redox instability [42].

### 3.2. Color Assessment

The color parameters (*L**, *a**, *b**) of minced pork samples showed dynamic changes over storage time, with significant shifts mainly associated with storage duration rather than treatment type (Table 2). The *L** values (lightness) generally increased from day 0 to day 9, particularly in sample T1 and the control sample, suggesting slight surface lightening during storage. However, no significant differences were noted between treated and control samples throughout the period, indicating that the antibacterial mixtures did not markedly affect surface lightness. Redness (*a** values) increased slightly by day 3 in all samples, likely due to oxymyoglobin formation during the initial storage, followed by stabilization or a minor decline, consistent with known myoglobin oxidation pathways under chilled conditions [40]. The sample with the T2 addition mixture showed slightly lower *a** values on day 9, though the differences remained statistically insignificant. Similarly, *b** values (yellowness) rose somewhat in all samples over time, likely reflecting pigment oxidation or mild lipid degradation, as reported in other studies on fresh and cured meats [43]. The lack of significant differences between treated and control samples across all color parameters is consistent with findings from Stadnik et al. (2022) [44], who reported that bioactive compounds from bacterial cultures did not negatively influence meat color. Similarly, Yang et al. (2023) [45] found that beef steaks treated with *Latilactobacillus sakei* RS-25 showed improved lightness and delayed color degradation during storage without adverse effects on general quality parameters. Similar results were reported by Incili et al. (2022) [46], who observed that applying postbiotics had no significant impact on chicken drumstick samples’ *L**, *a**, or *b** color parameters. These findings suggest that such treatments help maintain color stability, or at the very least do not negatively impact visual quality. While no significant improvement in color was observed, the results indicate that the supernatants had no adverse effect on color. This is favorable from a technological and consumer standpoint, as color stability plays a crucial role in meat acceptability at the point of sale [47,48].

### 3.3. Microbiological Quality

The microbial analysis of minced meat samples during 9 days of refrigerated storage revealed clear temporal patterns and treatment-related differences (Table 3, Appendix A). As expected, all samples exhibited a gradual increase in total viable counts (TVCs) throughout storage, characteristic of microbial proliferation under chilled conditions [49]. However, samples treated with antibacterial culture supernatants (mixtures T1 and T2) maintained significantly lower TVC levels compared to the untreated controls at all time points (*p* < 0.05). By day 9, the control sample reached 4.55 log CFU/g, while samples with addition mixtures T1 and T2 remained below 3.2 log CFU/g, suggesting that the added supernatants effectively delayed microbial proliferation. The image of TVC agar plates after incubation at a given temperature and the time for the three tested samples from one of the storage periods is presented in Appendix A. Lactic acid bacteria (LAB) counts followed a similar trend. By the end of storage, LAB counts in the control group had significantly increased to 4.26 log CFU/g (*p* < 0.05), whereas in samples treated with mixtures T1 and T2, levels remained below 2.5 log CFU/g. Enterobacteriaceae and *Escherichia coli* counts exhibited comparable behavior. While all samples steadily increased over time, the control consistently displayed higher levels, especially by day 9. In contrast, treated samples showed significantly restricted growth of these Gram-negative bacteria, with Enterobacteriaceae not exceeding 3.01 log CFU/g and *E. coli* remaining below the detection limit of 2 log CFU/g. Coagulase-positive staphylococci levels remained below the detection limit of 2 log CFU/g across all treatments and time points. Notably, no presence of *L. monocytogenes* or *Salmonella* spp. was detected in any sample throughout the study. This confirms the hygienic safety of the initial meat batches and suggests that the storage conditions were not conducive to the proliferation of these pathogens.

Although LAB are commonly present in meat and can contribute to spoilage under extended refrigeration [9], their growth was more gradual in treated samples. These results suggest that the supernatants not only suppressed spoilage organisms but also modulated the growth of LAB, potentially delaying quality loss [8]. Although Enterobacteriaceae are typically present in low numbers on fresh, intact meat, their spoilage potential is substantial due to their ability to produce discoloration and undesirable sulfurous or putrid odors [50]. These findings are in line with previous studies, such as Xu et al. (2023) [50], who reported significant reductions in Enterobacteriaceae and *Pseudomonas* spp. in vacuum-packed ground beef treated with mixed protective cultures containing *Lactilactobacillus sakei* and *Staphylococcus carnosus*. Similarly, Trabelsi et al. (2019) [3] observed a decrease in Enterobacteriaceae in probiotic-treated vacuum-packed beef, further supporting the antimicrobial potential of protective cultures against spoilage-associated Gram-negative microorganisms. In addition to *Listeria monocytogenes* and members of the *Enterobacteriaceae* family, other microorganisms, such as *Brochothrix thermosphacta* and *Clostridium* spp., also represent significant concerns regarding the microbiological quality of meat. Although these species were not included in the present analysis due to the defined scope of the study, their relevance to meat spoilage and public health is well recognized and will be addressed in future, more comprehensive investigations. 

The antimicrobial efficacy of the LAB and AAB supernatants used in pairs in this study is likely attributed to their production of organic acids (e.g., lactic, gluconic, and acetic acids), hydrogen peroxide, and bacteriocin-like compounds. These metabolites acidify the environment and disrupt microbial cell membrane integrity, which does not necessarily have to result in a statistically significant change in the pH value of the entire meat matrix. This phenomenon could reduce the viability of spoilage and pathogenic microorganisms over the storage period. Organic acids create an unfavorable environment by lowering the intracellular pH and interfering with nutrient transport across bacterial membranes [51,52]. Moreover, hydrogen peroxide exhibits oxidative activity against microbial membranes, enhancing the antimicrobial spectrum. The combined effects of LAB and AAB supernatants observed in this study may thus provide a synergistic barrier against spoilage bacteria, as supported by previous research evaluating fermented antimicrobial metabolites in meat systems [17,19,53].

In summary, the antibacterial mixtures demonstrated strong bioprotective potential by effectively inhibiting the growth of spoilage microorganisms throughout refrigerated storage. Their application could represent a promising strategy with which to enhance microbiological safety and extend the shelf life of fresh minced meat products.

### 3.4. TBARS and Lipid Oxidation Measurements

The TBARS assay was also employed to evaluate lipid peroxidation levels in minced meat samples over a 9-day refrigerated storage period (Table 4). Lipid oxidation is a critical factor affecting meat quality, leading to rancidity and off-flavors, thereby diminishing consumer acceptability [43]. By day 3, a significant increase (*p* < 0.05) in TBARS values was observed across all samples, signifying the initiation of lipid oxidation processes. This trend aligns with findings by Kaczmarek et al. (2021), who reported a similar early rise in TBARS values in ground pork during chilled storage [43]. This increase can be attributed to the susceptibility of unsaturated fatty acids to oxidative reactions upon exposure to oxygen and pro-oxidant factors inherent in meat matrices [43]. Between days 3 and 6, TBARS values continued to rise, albeit at a reduced rate, suggesting a progression in lipid oxidation. This phase may involve accumulating secondary oxidation products, such as aldehydes and ketones, contributing to off-flavors and odors in meat products [54]. Interestingly, by day 9, a decline in TBARS values was noted in all samples. This reduction could be due to the decomposition of malondialdehyde (MDA) or its interaction with other meat components, leading to less detectable free MDA. Such interactions may involve MDA binding to proteins or participation in further chemical reactions, resulting in less measurable MDA [55]. Samples treated with antibacterial supernatants (T1 and T2) consistently exhibited lower TBARS values than the control throughout the storage period. This observation suggests that compounds in the supernatants may exert antioxidative effects, possibly by scavenging free radicals or chelating metal ions that catalyze lipid oxidation [44]. These findings are consistent with Stadnik et al. (2022) [44], who demonstrated that lactic acid bacteria fermentation can enhance oxidative stability in dry-cured meats.

Table 4 also shows lipid stability in raw pork meat on days 0 and 9. All treatments significantly reduced cholesterol content after 9 days of storage (*p* < 0.05). The control sample demonstrated the smallest decrease in cholesterol levels (43.50 ± 1.11 mg/100 g), compared to the fall of 41.80 ± 0.400 mg/100 g in T1 and 40.22 ± 0.04 mg/100 g in T2. This suggests a potential antioxidative role of the antibacterial mixtures, contributing to reduced cholesterol oxidation. Cholesterol degradation is known to be influenced by both oxidative stress and microbial enzymatic activity, particularly under prolonged storage, and reductions in its content may reflect lipid peroxidation processes [56].

Peroxide values (PVs), indicators of primary lipid oxidation, increased significantly in all samples over time (*p* < 0.05). On day 9, sample T1 showed the highest PV (5.10 ± 0.41 mg/100 g), followed by T2 (2.35 ± 0.32 mg/100 g), while the control (C) had the lowest (1.29 ± 0.24 mg/100 g). Although T1 and T2 showed higher PVs than the control, this may be due to the early-stage accumulation of oxidation products or initial interactions between reactive components in the treatments and lipid substrates. This apparent rise in PV does not necessarily negate antioxidant potential, as further degradation or stabilization might occur at later stages. Additional studies are required to better understand the long-term oxidative behavior of these treatments.

Acid values (ACs), markers of hydrolytic rancidity, increased slightly in all samples without significant differences at day 9 (*p* > 0.05). The similar ACs in T1, T2, and the control suggest that the antibacterial mixtures did not accelerate lipid hydrolysis, aligning with Arrioja et al. (2020), who reported similar protective effects in beef [53]. While acid values increased in both the control (C) and T1 samples by day 9, a decrease was observed in the T2 sample, indicating that the specific antibacterial mixture used in T2 may have been more effective in limiting hydrolytic rancidity than T1. To better understand the underlying mechanisms, it is essential to consider that lipid degradation in meat during storage involves both oxidative and hydrolytic processes, which may be inversely modulated depending on treatment [57]. PV reflects early lipid oxidation, while AV indicates the hydrolysis of triglyceride into free fatty acids [57].

The parallel suppression of microbial growth and lipid oxidation observed in this study supports the multifunctional role of LAB and AAB-derived supernatants as biopreservative agents. Similar dual effects were reported by Papadochristopoulos et al. (2021) [58], who found that natural antimicrobials not only reduced microbial load in raw meat but also limited lipid peroxidation, leading to improved shelf life. Yu et al. (2021) [59] also demonstrated that bioactive compounds from microbial fermentation, such as organic acids and bacteriocins, exerted both bacteriostatic and antioxidative properties in pork sausages. Likewise, Domínguez et al. (2019) [56] and Wu et al. (2022) [60] observed a correlation between lower microbial counts and reduced TBARS values in meat systems treated with natural preservatives, suggesting the existence of a mechanistic link where microbial inhibition reduces the enzymatic and oxidative degradation of lipids. This relationship is particularly relevant given that some spoilage microorganisms (e.g., Enterobacteriaceae) produce pro-oxidative metabolites and enzymes that can accelerate lipid degradation. Therefore, the combined reduction in both microbial load and lipid oxidation in treated samples may indicate a synergistic preservation effect, where the inhibition of microbial metabolism also indirectly enhances oxidative stability. These findings reinforce the value of multi-target preservation strategies in extending the quality and safety of chilled meat products.

### 3.5. Fatty Acid Profile

The fatty acid composition of minced meat samples treated with antimicrobial mixtures (T1 and T2) and a control (C) was analyzed at two time points—on day 0 and day 9—of refrigerated storage at 4 ± 1 °C (Table 5). The main fatty acid classes analyzed included saturated (SFAs), monounsaturated (MUFAs), and polyunsaturated fatty acids (PUFAs). Saturated fatty acids were the most abundant, particularly palmitic (C16:0) and stearic acid (C18:0). Their levels remained relatively stable across all treatments, with minor increases observed on day 9. These trends are consistent with findings by Coombs et al. (2018) [61], suggesting that storage-related lipid oxidation had a minimal impact on SFA stability in lamb meat. In the MUFA category, oleic acid (C18:1cis9) was dominant, starting at 39% and remaining stable or decreasing slightly in treated samples (T1 and T2). At the same time, a marginal increase was noted in the control. This stability aligns with the results that Horbańczuk et al. (2019) [62] reported, indicating that vacuum packaging may help to preserve MUFA levels during short-term chilled storage. PUFAs, particularly linoleic (C18:2) and α-linolenic acid (C18:3n3), showed slight decreases over time, which were more pronounced in the control. After 9 days, C18:2 dropped by 0.5% across all treatments. Still, the most minor reduction was observed in samples T1 and T2, suggesting that the antimicrobial mixtures may offer mild protective effects against oxidative degradation [56]. Similarly, α-linolenic acid remained relatively stable in the treated groups, whereas a slight decrease was recorded in the control. This preservation is consistent with the previous work by Okoń et al. (2025) [15], who reported that natural antimicrobial treatments could delay PUFA oxidation in meat products. Adding antimicrobial mixtures T1 and T2 did not adversely affect the fatty acid composition and may have contributed to the better preservation of PUFAs. These findings suggest the potential of such treatments to maintain lipid quality in vacuum-packed minced meat during chilled storage.

### 3.6. Exploratory Factor Analysis (EFA)

Exploratory factor analysis (EFA) was conducted to identify latent structures that explain the variability among meat quality parameters in samples treated with a mixture of antimicrobial agents.

On day 0 (Table 6 and Table 7, Appendix A), three factors were extracted, explaining a cumulative variance of 75.38%. The first factor, “Microbiology” (32.88%), included ENT, TVC, LAB, EC, *b**, and STA, reflecting microbial activity and associated changes in yellow color. The second factor, “Oxidation–Color” (27.58%), comprised ORP, TBARS, *a**, *L**, *b**, and pH, indicating oxidative processes and color parameters. The third factor, “Lipids” (14.92%), included cholesterol, acid value, and peroxide value, representing lipid degradation. A weak negative correlation was observed between microbiological counts and the “Oxidation–Color” factor (r = −0.025), while lipid degradation showed positive correlations with both microbiological counts (r = 0.147) and “Oxidation–Color” (r = 0.167).

A comparable factor structure was observed on day 9 (Table 8 and Table 9, Appendix A), explaining 80.17% of the total variance. The “Microbiology” factor (39.79%) consisted of TVC, LAB, ENT, EC, and STA. The “Color–Oxidation” factor (28.62%) comprised pH, TBARS, ORP, *L**, *a**, and *b**. The “Lipids” factor (11.77%) included cholesterol, acid value, and peroxide value. On day 9, microbial load positively correlated with the “Color–Oxidation” factor (r = 0.139), whereas lipid degradation was weakly and negatively correlated with both microbial load (r = −0.039) and “Color–Oxidation” (r = −0.128). The factor structures obtained reflect the multifaceted nature of meat quality deterioration during storage.

The dominance of the “Microbiology” factor at both time points aligns with previous research, emphasizing microbial growth as a primary determinant of spoilage in stored meat [63]. The increasing influence of this factor over time suggests that microbial proliferation becomes more critical in shaping overall quality as storage progresses. The “Oxidation–Color” factor captures the interplay between oxidative degradation and visual quality, which are closely linked to consumer perception. The association between microbial activity and increased oxidation, especially on day 9, supports findings that microbial metabolism may accelerate lipid oxidation and pigment degradation [64]. Interestingly, the “Lipids” factor showed only modest contributions to the overall variance but revealed meaningful interactions with the other two factors. The weak negative correlation between lipid degradation and microbial or oxidative parameters could suggest that antimicrobial treatments provide some protection against lipid oxidation, consistent with prior studies highlighting the synergistic effects of antimicrobial and antioxidant agents in meat preservation [65,66]. These findings emphasize the interdependent roles of microbial activity, oxidative stability, and lipid degradation in determining meat quality. Antimicrobial agents appear to modulate these interactions, potentially delaying spoilage and preserving sensory attributes during storage [67,68].

### 3.7. Functional Principal Component Analysis (FPCA)

In this study, ready-to-cook minced meat samples were subjected to functional principal component analysis (FPCA) to investigate the effect of storage time on various parameters, including pH, ORP, color, and microbiological quality. For this purpose, measurements were collected at 0, 3, 6, and 9 days of refrigerated storage. To the best of our knowledge, the use of FPCA for the dynamic evaluation of meat quality during storage has been limited, underscoring the innovative nature of the present study.

The scree plot analysis (Figure 2) indicated that two functional principal components were sufficient to capture approximately 95.90% of the total variance observed in the dataset. The first component (PC1) explained 92.44% of the total variance, while the second component (PC2) accounted for an additional 3.46%.

PC1 mainly reflected the overall mean levels of the studied parameters across the storage period. The FPCA shows that higher PC1 values indicated above-average quality characteristics and were observed between the 4th and 6th day of storage. The analysis of PC1 values demonstrated that samples associated with the T2 treatment and samples related to the T1 treatment located in the upper right corner had the highest scores, suggesting the superior maintenance of quality attributes. Conversely, the control group samples displayed lower PC1 values, indicating faster deterioration in quality. Although PC2 accounted for a small proportion of the total variance (3.46%), it appeared to reflect differences in the timing of quality parameter changes. Positive PC2 values suggested that the samples were approaching or had passed their maximum quality peak, typically observed between days 4th and 6th, while negative values indicated that the samples were at their peak quality state. 

The biplot of PC1 and PC2 further confirmed that the samples treated with T2 and T1 generally maintained better quality characteristics for a longer duration than the control group (Figure 3).

Among the tested variants, the T2 treatment was the most effective in preserving meat quality over the storage period, followed by T1, while the untreated control samples exhibited the most rapid decline in quality parameters. These findings align with previous studies employing principal component analysis (PCA) to assess meat quality. For instance, Cañeque et al. (2004) utilized PCA to evaluate carcass and meat quality in light lambs, identifying key variables such as pH, color, and water-holding capacity as significant contributors to meat quality differentiation [69]. Similarly, Hu et al. (2007) applied PCA to pork quality assessment, highlighting the importance of pH, color, moisture, and intramuscular fat in distinguishing quality traits [70]. While these studies focused on static measurements, the current research extends the application by incorporating the temporal aspect through FPCA, offering a more dynamic understanding of quality changes during storage. The superior performance of the T2 treatment in preserving meat quality over time suggests its potential efficacy as a preservative agent. The ability of FPCA to effectively capture and differentiate the temporal quality trajectories among treatments underscores its utility in meat science research. The insights gained through FPCA could aid in developing improved preservation strategies and quality monitoring systems for the meat industry, ultimately extending shelf life and reducing food waste. Future studies could further explore the application of FPCA in conjunction with other analytical techniques to enhance the monitoring and prediction of meat quality changes during storage [71].

## 4. Conclusions

This study confirmed that cell-free supernatants derived from *Lacticaseibacillus paracasei* B1, *Lactiplantibacillus plantarum* O24, *Gluconobacter oxydans* KNS32, and *Komagataeibacter saccharivorans* KOM1 can serve as effective biopreservative agents for ready-to-cook minced pork. Their application enhances the following:(1)Microbiological safety by inhibiting spoilage-related microbial growth;(2)Oxidative stability, through a reduction in lipid oxidation;(3)Technological quality, by maintaining parameters such as pH and color during refrigerated storage.

The synergistic use of lactic acid bacteria (LAB) and acetic acid bacteria (AAB) preparations presents a novel, natural alternative to synthetic preservatives, aligning with current consumer demand for clean-label food products.

Among the tested formulations, the mixture containing *Lactiplantibacillus plantarum* O24 and *Gluconobacter oxydans* KNS32 proved particularly effective in slowing down spoilage processes. Multivariate analyses, including exploratory factor analysis (EFA) and functional principal component analysis (FPCA), indicated that microbial growth control was the key to overall quality preservation. At the same time, oxidative and color parameters acted as secondary quality indicators. The findings emphasize the importance of microbiological stability in extending shelf life, particularly in minimally processed meat products vulnerable to rapid spoilage.

While promising, the study was conducted under controlled laboratory conditions and did not simulate real-world factors such as temperature variation or transport. We believe that the research should be continued to expand the range of possible solutions and conditions, and include more food samples.

Future research should address the following factors:(1)Sensory evaluation (flavor, aroma, texture) and consumer acceptance;(2)Validation under industrial conditions, including logistical stressors;(3)Broader formulation testing to confirm scalability and robustness.

Given their demonstrated antimicrobial activity, ease of preparation, and cell-free nature, these supernatants offer a feasible and scalable approach for commercial application in meat processing environments. In conclusion, this study highlights the synergistic potential of LAB and AAB preparations as innovative natural preservatives for minimally processed meat products, providing a strong basis for further development toward industrial application.

## Figures and Tables

**Figure 1 foods-14-01934-f001:**
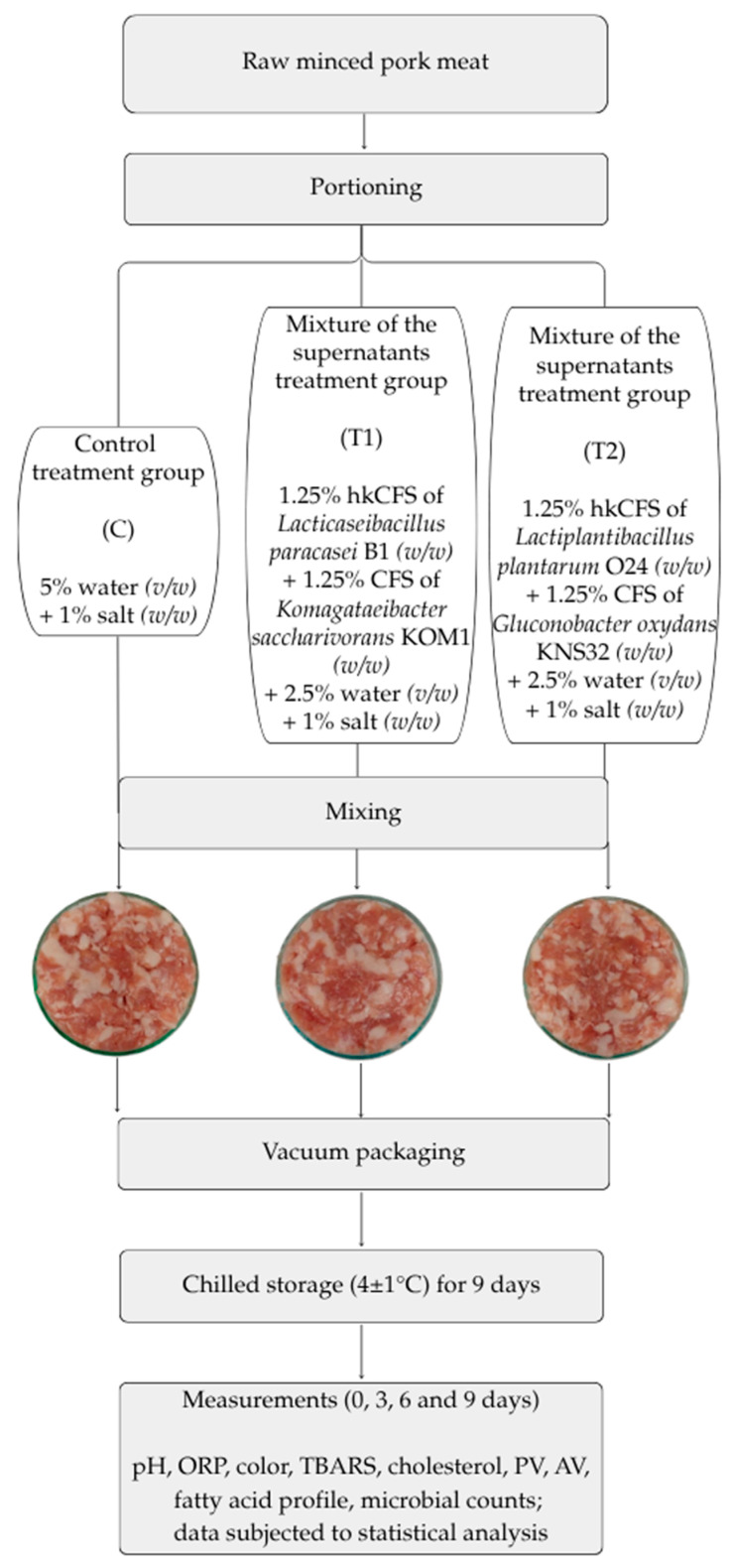
Illustration of experimental procedure and photos of tested products after mixing. ORP—oxidation reduction potential; TBARS—thiobarbituric acid reactive substance; PV—peroxide value; AV—acid value.

**Figure 2 foods-14-01934-f002:**
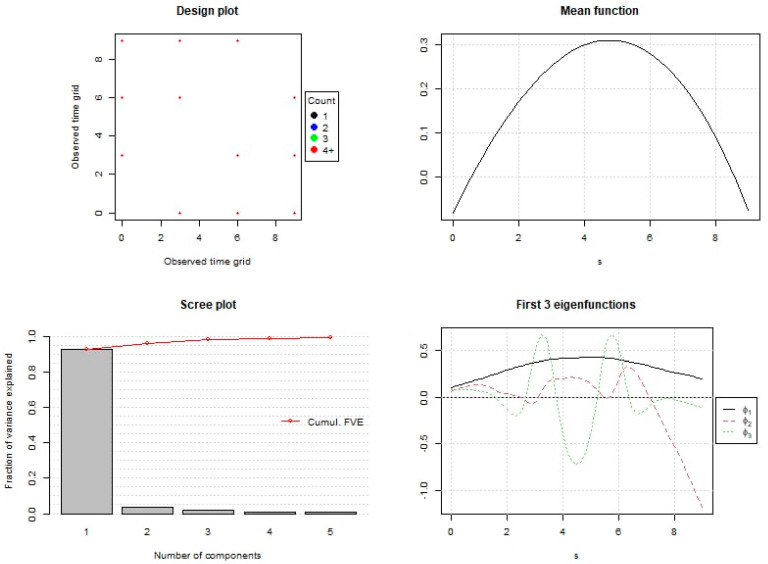
Summary of functional principal component analysis. Different colors represent different numbers of observations, but in this case, only red points are shown because all observations are concentrated on dates with sample size ≥ 4. No points are shown for dates with no observations.

**Figure 3 foods-14-01934-f003:**
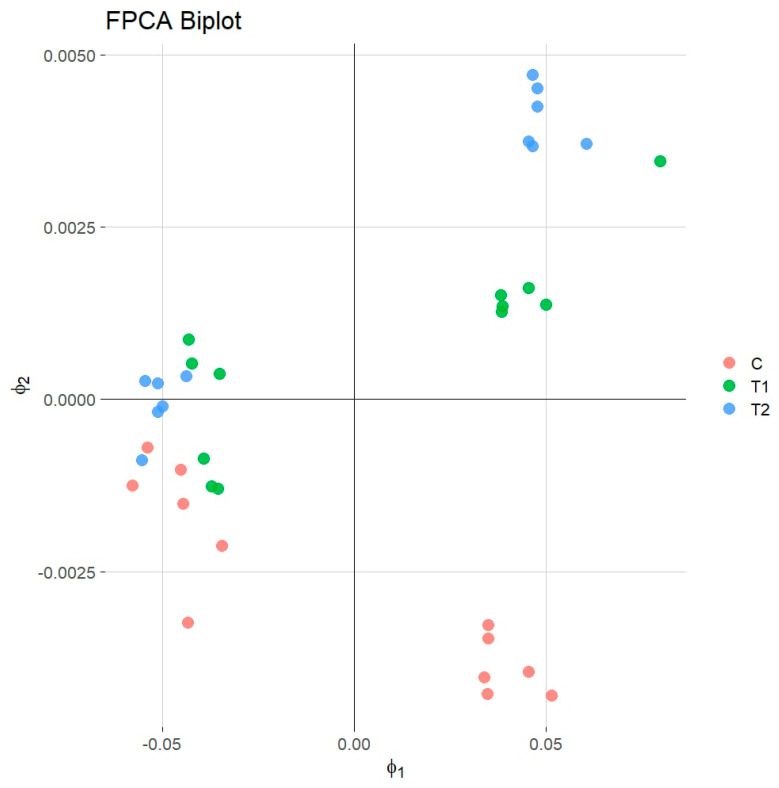
Biplot functional principal component analysis.

**Table 1 foods-14-01934-t001:** pH changes in ready-to-cook minced pork samples during storage.

Parameter	Time [Days]	Treatment		
		T1	T2	C
pH	0	6.02 ± 0.03 ^aA^	6.00 ± 0.04 ^aA^	6.03 ± 0.07 ^aA^
	3	5.91 ± 0.04 ^bA^	5.91 ± 0.06 ^bA^	5.91 ± 0.02 ^bA^
	6	5.54 ± 0.04 ^cA^	5.38 ± 0.02 ^cB^	5.65 ± 0.10 ^cA^
	9	5.56 ± 0.09 ^cA^	5.55 ± 0.02 ^cA^	5.54 ± 0.10 ^cA^
ORP (mV)	0	333.50 ± 9.63 ^aA^	331.50 ± 8.00 ^aA^	335.50 ± 11.50 ^aA^
	3	336.50 ± 7.88 ^aA^	334.00 ± 9.63 ^aA^	339.50 ± 12.75 ^abA^
	6	358.00 ± 21.63 ^bA^	362.00 ± 29.50 ^bA^	348.00 ± 15.38 ^bA^
	9	349.50 ± 16.25 ^cA^	336.50 ± 9.88 ^cA^	374.50 ± 32.75 ^cA^

The values are expressed as median ± Q. ^a,b,c^—the median, marked with lowercase letters, differs significantly between times (*p* < 0.05). ^A,B^—the median, marked with uppercase letters, differs significantly between treatments (*p* < 0.05).

**Table 2 foods-14-01934-t002:** Color changes in ready-to-cook minced pork samples during storage.

Color Parameters	Time [Days]	Treatment		
		T1	T2	C
*L**	0	56.37 ± 0.85 ^aA^	57.89 ± 1.18 ^aA^	56.74 ± 0.59 ^aA^
	3	56.10 ± 1.24 ^bA^	56.46 ± 1.98 ^bA^	55.88 ± 1.39 ^aA^
	6	58.44 ± 0.50 ^cA^	56.92 ± 0.59 ^cAB^	56.49 ± 0.77 ^aB^
	9	58.07 ± 0.51 ^dA^	57.45 ± 0.50 ^acA^	57.81 ± 1.02 ^bA^
*a**	0	3.96 ± 0.35 ^aA^	4.05 ± 0.36 ^aA^	4.69 ± 0.77 ^aA^
	3	6.61 ± 0.33 ^bA^	6.41 ± 0.72 ^bAB^	5.83 ± 0.34 ^bB^
	6	6.12 ± 0.32 ^cA^	6.46 ± 0.29 ^bA^	5.90 ± 0.41 ^cB^
	9	6.13 ± 0.18 ^cA^	5.45 ± 0.64 ^cA^	6.01 ± 0.32 ^cA^
*b**	0	9.15 ± 0.16 ^aA^	9.25 ± 0.36 ^aA^	8.16 ± 0.31 ^aB^
	3	9.40 ± 0.28 ^bA^	9.15 ± 0.25 ^bA^	8.43 ± 0.28 ^bB^
	6	10.19 ± 0.26 ^cA^	10.25 ± 0.28 ^cA^	9.32 ± 0.23 ^cB^
	9	9.87 ± 0.26 ^dA^	10.06 ± 0.12 ^dA^	9.72 ± 0.42 ^dA^

*L***a***b**—color parameters. The values are expressed as median ± Q. ^a,b,c,d^—the median marked with lowercase letters differs significantly between times (*p* < 0.05); ^A,B^—the median marked with uppercase letters differs significantly between treatments (*p* < 0.05).

**Table 3 foods-14-01934-t003:** Microbial quality in ready-to-cook minced pork samples during storage.

Analyses	Time [Days]	Treatment		
		T1	T2	C
TVC (log CFU/g)	0	2.48 ± 0.15 ^aA^	2.43 ± 0.10 ^aB^	3.03 ± 0.12 ^aC^
	3	2.50 ± 0.12 ^bA^	2.63 ± 0.17 ^bB^	2.92 ± 0.17 ^bC^
	6	2.80 ± 0.19 ^cA^	2.95 ± 0.15 ^cB^	3.28 ± 0.15 ^cC^
	9	2.98 ± 0.11 ^dA^	3.11 ± 0.19 ^dB^	4.55 ± 0.12 ^dC^
LAB (log CFU/g)	0	1.52 ± 0.15 ^aA^	1.31 ± 0.15 ^aB^	2.05 ± 0.11 ^aC^
	3	1.52 ± 0.12 ^abA^	1.93 ± 0.14 ^bB^	1.44 ± 0.19 ^AC^
	6	1.95 ± 0.13 ^cA^	1.91 ± 0.16 ^cB^	2.40 ± 0.18 ^cC^
	9	2.45 ± 0.15 ^dA^	2.34 ± 0.11 ^dB^	4.26 ± 0.14 ^dC^
ENT (log CFU/g)	0	2.31 ± 0.107 ^aA^	2.28 ± 0.20 ^aB^	2.69 ± 0.17 ^aC^
	3	2.29 ± 0.132 ^bA^	2.46 ± 0.16 ^bB^	2.78 ± 0.14 ^bC^
	6	2.52 ± 0.16 ^cA^	2.84 ± 0.13 ^cB^	3.13 ± 0.14 ^cC^
	9	2.83 ± 0.19 ^dA^	3.01 ± 0.13 ^dB^	4.01 ± 0.17 ^dC^
EC (log CFU/g)	0	<2.00	<2.00	2.13 ± 0.13 ^aC^
	3	<2.00	<2.00	2.11 ± 0.12 ^bC^
	6	2.11 ± 0.11 ^cA^	2.18 ± 0.14 ^cB^	2.41 ± 0.11 ^cC^
	9	<2.00	<2.00	<2.00
STA (log CFU/g)	0	<2.00	<2.00	<2.00
	3	<2.00	<2.00	<2.00
	6	<2.00	<2.00	<2.00
	9	<2.00	<2.00	<2.00
LM (log CFU/g)	0	nd	nd	nd
	3	nd	nd	nd
	6	nd	nd	nd
	9	nd	nd	nd
SAL (log CFU/g)	0	nd	nd	nd
	3	nd	nd	nd
	6	nd	nd	nd
	9	nd	nd	nd

Explanatory notes: TVC—total viable count; LAB—lactic acid bacteria; ENT—bacteria Enterobacteriaceae family; EC—*Escherichia coli*; STA—coagulase-positive staphylococci (Staphylococcus aureus and other species); LM—*Listeria monocytogenes*; SAL—*Salmonella* spp.; nd—not detected; CFU—colony-forming unit. The values are expressed as median ± Q. ^a,b,c,d^ —the median marked with lowercase letters differ significantly between times (*p* < 0.05); ^A,B,C^—the median marked with uppercase letters differ significantly between treatments (*p* < 0.05).

**Table 4 foods-14-01934-t004:** TBARS and lipid stability parameters in ready-to-cook minced pork samples during storage.

Parameter	Time	Treatment		
		T1	T2	C
TBARS (mg MDA/kg of product)	0	0.52 ± 0.13 ^aA^	0.46 ± 0.13 ^aA^	0.49 ± 0.09 ^aA^
	3	0.69 ± 0.17 ^bA^	0.69 ± 0.19 ^bA^	0.70 ± 0.21 ^bA^
	6	0.40 ± 0.14 ^cA^	0.45 ± 0.10 ^cA^	0.40 ± 0.09 ^cA^
	9	0.34 ± 0.05 ^dA^	0.37 ± 0.03 ^cA^	0.43 ± 0.09 ^dA^
Cholesterol (mg/100 g)	0.	53.22 ± 1.03 ^aA^	53.95 ± 1.45 ^aA^	53.42 ± 0.35 ^aA^
	9	41.80 ± 0.40 ^bA^	40.22 ± 0.04 ^bB^	43.50 ± 1.11 ^bC^
Peroxide value (mg/100 g)	0	0.55 ± 0.10 ^aA^	0.62 ± 0.09 ^aA^	0.94 ± 0.18 ^aA^
	9	5.10 ± 0.41 ^bA^	2.35 ± 0.32 ^bAB^	1.29 ± 0.24 ^aB^
Acid value (mg/100 g)	0	0.91 ± 0.06 ^aA^	1.35 ± 0.15 ^aA^	0.70 ± 0.04 ^aA^
	9	1.20 ± 0.05 ^bA^	1.11 ± 0.09 ^aA^	1.01 ± 0.07 ^bA^

TBARS—thiobarbituric acid reactive substance; MDA—malondialdehyde. The values are expressed as median ± Q. ^a,b,c,d^—the median values, marked with lowercase letters, differ significantly between times (*p* < 0.05). ^A,B,C^—the median marked with uppercase letters differ significantly between treatments (*p* < 0.05).

**Table 5 foods-14-01934-t005:** Individual fatty acid content [g/100 g total fatty acid] in ready-to-cook minced pork samples during storage.

Category	Fatty Acids		Time [Days]	Treatment		
				T1	T2	C
SFA	Capric acid	10:0	0	0.10 ± 0.00 ^aA^	0.10 ± 0.00 ^aA^	0.10 ± 0.00 ^aA^
			9	0.10 ± 0.00 ^aA^	0.10 ± 0.00 ^aA^	0.10 ± 0.00 ^aA^
SFA	Lauric acid	12:0	0	0.10 ± 0.00 ^aA^	0.10 ± 0.00 ^aA^	0.10 ± 0.00 ^aA^
			9	0.10 ± 0.00 ^aA^	0.10 ± 0.00 ^aA^	0.10 ± 0.00 ^aA^
SFA	Myristic acid	14:0	0	1.25 ± 0.01 ^aA^	1.27 ± 0.02 ^aAB^	1.32 ± 0.01 ^aC^
			9	1.28 ± 0.01 ^bA^	1.27 ± 0.02 ^aA^	1.28 ± 0.01 ^bA^
SFA	Pentadecanoic acid	15:0	0	0.10 ± 0.00 ^aA^	0.10 ± 0.00 ^aA^	0.10 ± 0.00 ^aA^
			9	0.10 ± 0.00 ^aA^	0.10 ± 0.00 ^aA^	0.10 ± 0.00 ^aA^
SFA	Palmitic acid	16:0	0	23.42 ± 0.11 ^aA^	23.35 ± 0.14 ^aA^	23.55 ± 0.11 ^aA^
			9	23.58 ± 0.10 ^bA^	23.53 ± 0.14 ^aA^	23.58 ± 0.11 ^bA^
MUFA	Palmitoleic acid	16:1	0	2.70 ± 0.01 ^aA^	2.62 ± 0.04 ^aA^	2.65 ± 0.03 ^aA^
			9	2.68 ± 0.01 ^bA^	2.65 ± 0.03 ^bA^	2.73 ± 0.03 ^bA^
SFA	Heptadecanoic acid	17:0	0	0.35 ± 0.01 ^aA^	0.35 ± 0.01 ^aA^	0.33 ± 0.02 ^aA^
			9	0.35 ± 0.01 ^aA^	0.32 ± 0.01 ^bA^	0.31 ± 0.01 ^bA^
MUFA	Heptadecenoic acid	17:1	0	0.30 ± 0.00 ^aA^	0.30 ± 0.00 ^aA^	0.30 ± 0.00 ^aA^
			9	0.30 ± 0.00 ^aA^	0.30 ± 0.00 ^aA^	0.30 ± 0.00 ^aA^
SFA	Stearic acid	18:0	0	12.52 ± 0.03 ^aA^	12.52 ± 0.10 ^aA^	12.65 ± 0.03 ^aA^
			9	12.83 ± 0.14 ^aA^	12.78 ± 0.11 ^bA^	12.67 ± 0.06 ^bA^
MUFA	Elaidic acid (trans isomer of oleic)	18:1trans	0	0.20 ± 0.00 ^aA^	0.20 ± 0.00 ^aA^	0.20 ± 0.00 ^aA^
			9	0.20 ± 0.00 ^aA^	0.20 ± 0.00 ^aA^	0.20 ± 0.00 ^aA^
MUFA	Oleic acid	18:1cis9	0	39.03 ± 0.18 ^aA^	39.34 ± 0.10 ^aAB^	38.70 ± 0.03 ^aAC^
			9	38.83 ± 0.11 ^aA^	39.07 ± 0.04 ^bA^	39.22 ± 0.02 ^bA^
MUFA	Vaccenic acid	18:1cis11	0	2.97 ± 0.02 ^aA^	2.93 ± 0.02 ^aA^	2.90 ± 0.01 ^aA^
			9	2.95 ± 0.01 ^aA^	2.85 ± 0.03 ^aA^	3.00 ± 0.01 ^aA^
MUFA	Other positional isomers of oleic acid	18:1 c other	0	0.23 ± 0.02 ^aA^	0.22 ± 0.01 ^aA^	0.23 ± 0.02 ^aA^
			9	0.22 ± 0.01 ^bA^	0.22 ± 0.09 ^aA^	0.23 ± 0.02 ^aA^
PUFA	Linoleic acid (n-6)	18:2	0	13.05 ± 0.03 ^aA^	13.41 ± 0.16 ^aA^	13.17 ± 0.11 ^aA^
			9	12.83 ± 0.12 ^aA^	12.87 ± 0.21 ^bA^	12.58 ± 0.16 ^aA^
PUFA	Alpha-linolenic acid (ALA, n-3)	18:3 n3	0	0.92 ± 0.03 ^aA^	0.95 ± 0.03 ^aA^	0.95 ± 0.03 ^aA^
			9	0.93 ± 0.05 ^aA^	0.95 ± 0.06 ^aA^	0.90 ± 0.05 ^bA^
PUFA	Conjugated linoleic acid (CLA, PUFA variant)	18:2c9t11	0	0.10 ± 0.00 ^aA^	0.10 ± 0.00 ^aA^	0.10 ± 0.00 ^aA^
			9	0.10 ± 0.00 ^aA^	0.10 ± 0.00 ^aA^	0.10 ± 0.00 ^aA^
SFA	Arachidic acid	20:0	0	0.20 ± 0.00 ^aA^	0.20 ± 0.00 ^aA^	0.20 ± 0.00 ^aA^
			9	0.20 ± 0.00 ^aA^	0.20 ± 0.00 ^aA^	0.20 ± 0.00 ^aA^
MUFA	Gadoleic acid	20:1	0	0.90 ± 0.00 ^aA^	0.90 ± 0.00 ^aA^	0.90 ± 0.00 ^aA^
			9	0.90 ± 0.00 ^aA^	0.90 ± 0.00 ^aA^	0.90 ± 0.00 ^aA^
PUFA	Eicosadienoic acid	20:2	0	0.62 ± 0.01 ^aA^	0.633 ± 0.017 ^aBA^	0.58 ± 0.01 ^aC^
			9	0.58 ± 0.01 ^bA^	0.567 ± 0.017 ^bA^	0.57 ± 0.02 ^bA^
PUFA	Dihomo-γ-linolenic acid (DGLA, n-6)	20:3n6	0	0.10 ± 0.00 ^aA^	0.10 ± 0.00 ^aA^	0.10 ± 0.00 ^aA^
			9	0.10 ± 0.00 ^aA^	0.10 ± 0.00 ^aA^	0.10 ± 0.00 ^aA^
PUFA	Arachidonic acid (AA, n-6)	20:4n6	0	0.43 ± 0.03 ^aA^	0.37 ± 0.03 ^aA^	0.42 ± 0.06 ^aA^
			9	0.38 ± 0.01 ^aA^	0.40 ± 0.00 ^bB^	0.40 ± 0.00 ^aBC^
PUFA	Adrenic acid (n-6)	22:4n6	0	0.12 ± 0.01 ^aA^	0.12 ± 0.01 ^aA^	0.12 ± 0.01 ^aA^
			9	0.12 ± 0.01 ^bA^	0.12 ± 0.01 ^aA^	0.12 ± 0.01 ^bA^
PUFA	Docosapentaenoic acid (DPA, n-3)	22:5n3	0	0.10 ± 0.00 ^aA^	0.10 ± 0.00 ^aA^	0.10 ± 0.00 ^aA^
			9	0.10 ± 0.00 ^aA^	0.10 ± 0.00 ^aA^	0.10 ± 0.00 ^aA^

SFA—saturated fatty acids; MUFA—monounsaturated fatty acids; PUFA—polyunsaturated fatty acids. The values are expressed as median ± Q. ^a,b^—the median marked with lowercase letters differ significantly between times (*p* < 0.05). ^A,B,C^—the median marked with uppercase letters differ significantly between treatments (*p* < 0.05).

**Table 6 foods-14-01934-t006:** Factor loadings for day 0 of storage.

Components [Day 0]	Factors			Specificity Factor
	1	2	3	
ENT	1.005			−0.014
TVC	0.974			0.041
LAB	0.900			0.180
EC	0.779			0.384
*b**	−0.778	0.537		0.086
STA	−0.579			0.657
ORP		0.919		0.172
TBARS		0.902		0.187
*a**		0.826		0.202
*L**		0.787		0.266
pH		0.747		0.385
cholesterol			0.885	0.235
peroxide value			0.783	0.297
acid value			−0.717	0.368

ENT—Enterobacteriaceae; TVC—total viable count; STA—Staphylococcus aureus; LAB—lactic acid bacteria count; EC—Escherichia coli count; ORP—oxidative reduction potential; TBARS—thiobarbituric acid reactive substances; *L***a***b**—color parameters.

**Table 7 foods-14-01934-t007:** Level of variance explanation for day 0 of storage.

Factors	Total	% of Variance	Cumulative %
1	4.603	32.877	32.877
2	3.861	27.576	60.453
3	2.089	14.922	75.375

**Table 8 foods-14-01934-t008:** Factor loadings for day 9 of storage.

Components [Day 9]	Factors			Specificity Factor
	1	2	3	
ENT		−0.689		0.531
TVC		0.924		0.105
LAB		0.804		0.217
EC		0.922		0.040
*b**		0.753		0.445
STA		0.758		0.250
ORP	0.997			−0.004
TBARS	0.978			0.027
*a**	0.992			0.007
*L**	0.988			0.016
pH	0.931			0.112
cholesterol			0.664	0.362
peroxide value			0.727	0.235
acid value			0.682	0.434

ENT—Enterobacteriaceae; TVC—total viable count; STA—Staphylococcus aureus; LAB—lactic acid bacteria count; EC—Escherichia coli count; ORP—oxidative-reduction potential; TBARS—thiobarbituric acid reactive substances; *L***a***b**—color parameters.

**Table 9 foods-14-01934-t009:** Level of variance explanation for day 9 of storage.

Factors	Total	% of Variance	Cumulative %
1	5.570	39.787	39.787
2	4.006	28.615	68.402
3	1.648	11.768	80.170

## Data Availability

The original contributions presented in the study are included in the article/Appendix A, further inquiries can be directed to the corresponding authors.

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
