# Peer review of "Novel Applications of Lactic Acid and Acetic Acid Bacteria Preparations in Shaping the Technological and Microbiological Quality of Ready-to-Cook Minced Pork"

_foods, 2025, doi:10.3390/foods14111934_

Round 1
Reviewer 1 Report
Comments and Suggestions for Authors
This paper presents a study on the use of cell-free supernatants from lactic acid and acetic acid bacteria as natural biopreservatives for ready-to-cook minced pork. It evaluates their effects on microbial stability, physicochemical properties, and lipid oxidation during refrigerated storage. Based on the findings, I believe the manuscript is scientifically sound and requires only minor revisions.
In the paper you mention that the antimicrobial effects come from organic acids, hydrogen peroxide, and bacteriocin-like compounds, but it doesn’t specify how much of each is present. Without this kind of information, it’s difficult to know how consistent or reproducible the results would be if the experiment were repeated. Could slight differences in fermentation or culture conditions alter the composition significantly? What exactly is in the supernatants? How sure are we that the supernatants caused the microbial reduction?
The authors focused on E. coli, Listeria monocytogenes, Salmonella, and a few spoilage groups like Enterobacteriaceae. But what about other common meat spoilers like Brochothrix thermosphacta or Clostridium species? Including them could provide a more complete picture of the microbial stability of the product.
Are you reading too much into the FPCA? The first principal component explains more than 90% of the variation, which is great. But some of the conclusions seem to rely on changes in PC2, which explains just 3%. Are those subtle shifts really meaningful? Or could they just be noise?
How do we explain the peroxide values? I mean in treated samples (especially T1) the peroxide values were actually higher than in the control. That seems a little strange if we’re claiming antioxidant protection. Is it possible that these are early-stage oxidation products accumulating before further degradation? This should be discussed more clearly.
The study was done under controlled lab conditions: vacuum-sealed packs, consistent refrigeration, etc. But commercial meat distribution often involves variable transport times, handling, and temperature fluctuations. How realistic is it to expect the same level of performance from these treatments under real-world conditions? Can these results apply to real-world meat production?
Author Response
Response to Reviewer Comment 1:
In the paper you mention that the antimicrobial effects come from organic acids, hydrogen peroxide, and bacteriocin-like compounds, but it doesn’t specify how much of each is present. Without this kind of information, it’s difficult to know how consistent or reproducible the results would be if the experiment were repeated. Could slight differences in fermentation or culture conditions alter the composition significantly? What exactly is in the supernatants? How sure are we that the supernatants caused the microbial reduction?
We thank the Reviewer for this thoughtful and important comment. To clarify, the composition of the cell-free supernatants (CFS) was partially characterized in our study. As detailed in Section 2.1, the dominant organic acids present in the supernatants were quantified: lactic acid (approx. 8.3 mg/mL), acetic acid (approx. 6.7 mg/mL), and gluconic acid (approx. 5.9 mg/mL). While this analysis provides insight into the primary antimicrobial constituents, we fully agree that further quantification, particularly of hydrogen peroxide and bacteriocin-like metabolites, would enhance both the reproducibility and mechanistic interpretation of the findings.
To address the Reviewer’s concern regarding the potential impact of fermentation variability, an explanatory statement was added to the revised manuscript (Section 2.1) to clarify how consistency across preparations was ensured. The following sentence was included:
To address potential variability, bacterial cultures were standardized by employing defined media, controlled incubation parameters, and consistent cell density. The same preparation procedure was applied across all replicates. Additionally, heat-treated, cell-free supernatants derived from defined strains were used to ensure batch-to-batch consistency and to eliminate variability associated with ongoing microbial metabolism during storage.
This methodological standardization was implemented specifically to reduce the risk of compositional variability between replicates and batches, thereby increasing the reliability of the observed antimicrobial effects.
In addition, a supplementary analysis of protein content in the CFS was performed to explore the presence of bacteriocin-like substances. Preliminary findings confirmed the presence of proteinaceous compounds potentially linked to antimicrobial activity. These data are part of a follow-up study currently in preparation.
To strengthen the attribution of microbial reduction to the CFS, the following experimental safeguards were incorporated:
- All treatments were benchmarked against a water/salt-only control;
- Viable counts of key spoilage organisms (TVC, LAB, Enterobacteriaceae, E. coli) were significantly reduced in treated samples across all time points;
- Pathogens (Listeria monocytogenes, Salmonella spp.) remained undetectable throughout storage.
These combined measures support the conclusion that the microbial inhibition observed was directly attributable to the properties of the standardized, cell-free supernatants.
Response to Reviewer Comment 2:
The authors focused on E. coli, Listeria monocytogenes, Salmonella, and a few spoilage groups like Enterobacteriaceae. But what about other common meat spoilers like Brochothrix thermosphacta or Clostridium species? Including them could provide a more complete picture of the microbial stability of the product.
Thank you for this valuable and insightful suggestion. In this study, the microbial analysis focused on indicator organisms and pathogens most commonly used in meat safety assessments, including E. coli, Listeria monocytogenes, Salmonella spp., and the Enterobacteriaceae family. These groups serve as established benchmarks in both industry practice and regulatory frameworks for evaluating hygiene, spoilage potential, and public health risk.
In response to the Reviewer’s comment, a sentence has been added to the revised manuscript to acknowledge the relevance of other spoilage organisms and to clarify the scope of the current analysis. The following text was included:
In addition to Listeria monocytogenes and members of the Enterobacteriaceae family, other microorganisms, such as Brochothrix thermosphacta and Clostridium spp., also represent significant concerns regarding the microbiological quality of meat. Although these species were not included in the present analysis due to the defined scope of the study, their relevance to meat spoilage and public health is well recognized and will be addressed in future, more comprehensive investigations.
We agree that including these additional organisms would enhance the comprehensiveness of the microbial stability assessment. Therefore, future studies are planned to broaden the microbiological scope and provide deeper insight into spoilage dynamics in meat treated with LAB/AAB-derived supernatants.
Response to Reviewer Comment 3:
Are you reading too much into the FPCA? The first principal component explains more than 90% of the variation, which is great. But some of the conclusions seem to rely on changes in PC2, which explains just 3%. Are those subtle shifts really meaningful? Or could they just be noise?
We appreciate the reviewer’s observation regarding the limited variance explained by PC2. We agree that PC1 accounts for the vast majority of the variability and provides the primary basis for interpretation. Our mention of PC2 aimed to illustrate subtle temporal differences in sample quality trends rather than to draw strong conclusions. To address this, we have clarified the interpretation in the revised text and toned down any overstatements regarding PC2.
Revised text:
Although PC2 accounted for a small proportion of the total variance (3.46%), it appeared to reflect differences in the timing of quality parameter changes. Positive PC2 values suggested that the samples were approaching or had passed their maximum quality peak, typically observed between days 4 and 6, while negative values indicated that the samples were at their peak quality state.
Response to Reviewer Comment 4:
How do we explain the peroxide values? I mean in treated samples (especially T1) the peroxide values were actually higher than in the control. That seems a little strange if we’re claiming antioxidant protection. Is it possible that these are early-stage oxidation products accumulating before further degradation? This should be discussed more clearly.
Thank you for the insightful comment. We acknowledge that the higher peroxide values in treated samples, particularly T1, may seem contradictory to the expected antioxidant effect. We have revised the text to clarify that this may be due to early-stage oxidation or interactions between the treatment compounds and lipids, and we emphasize the need for further investigation.
Revised text:
Peroxide values (PV), indicators of primary lipid oxidation, increased significantly in all samples over time (p < 0.05). On day 9, sample T1 showed the highest PV (5.10 ± 0.41 mg/100 g), followed by T2 (2.35 ± 0.32 mg/100 g), while the control (C) had the lowest (1.29 ± 0.24 mg/100 g). Although T1 and T2 showed higher PVs than the control, this may be due to early-stage accumulation of oxidation products or initial interactions between reactive components in the treatments and lipid substrates. This apparent rise in PV does not necessarily negate antioxidant potential, as further degradation or stabilization might occur at later stages. Additional studies are required to better understand the long-term oxidative behavior of these treatments.
Response to Reviewer Comment 5:
The study was done under controlled lab conditions: vacuum-sealed packs, consistent refrigeration, etc. But commercial meat distribution often involves variable transport times, handling, and temperature fluctuations. How realistic is it to expect the same level of performance from these treatments under real-world conditions? Can these results apply to real-world meat production?
We agree that real-world meat distribution involves variable factors, such as fluctuating temperatures, inconsistent packaging, and longer transport times, that differ from our controlled experimental setup. Our study was designed as a foundational investigation to assess the intrinsic potential of LAB and AAB supernatants under standardized conditions, isolating their effects without the confounding impact of external variability. The use of vacuum-sealed packaging and cold storage mirrors best-practice protocols in high-quality processing facilities, particularly for ready-to-cook products.
However, we acknowledge that these conditions may not fully reflect lower-tier supply chains or retail-level storage. Notably, the oxidative and microbiological changes observed, even under optimal storage, highlight the delicate balance in meat preservation. The significant performance of treatment T2 under these conservative conditions suggests it may provide a buffer against mild deviations in real-world scenarios. Nonetheless, the protective mechanisms, such as inhibition of microbial respiration and reactive oxygen species scavenging, need to be tested under stress conditions (e.g., temperature abuse, aerobic packaging) to better assess their resilience. We see this study as a necessary first step, with real-world simulations as a logical extension to validate applicability across the broader meat industry.
Reviewer 2 Report
Comments and Suggestions for Authors
- While the overall English is acceptable, some sentences are overly long or complex, which can hinder clarity. Consider breaking long sentences into two or rephrasing to improve readability;
- There are instances of redundancy or slightly awkward phrasing (e.g., "this approach aligns with the broader development of functional foods" could be simplified for impact);
- The discussion section, though adequate, could benefit from a deeper comparison with previous findings in the literature. For example, highlight how the microbial activity and lipid oxidation trends observed in your study align or differ from related studies;
- Consider addressing potential limitations of your experimental design, such as the use of only two bacterial mixtures or limited sample sizes (n=3 per group), and how this might affect generalizability;
- The implications for industry application could be more explicitly addressed. How feasible is the use of these bacterial supernatants in commercial settings?
- In some tables, superscripts indicating statistical differences are not well explained in the captions. Ensure every table includes clear footnotes for interpretation;
- Although generally well-prepared, some tables are densely packed and could be made more reader-friendly with better spacing or simplified layouts;
- Ensure all abbreviations used in tables and figures (e.g., TBARS, ORP, FPCA) are defined either in the table/figure legend;
- The flow of the Results and Discussion sections could be improved by better separating findings from interpretations. Consider placing numerical data and findings first, then interpreting or comparing to literature in subsequent sentences.
- The Conclusions could be more concise and focused on the main practical takeaways, perhaps in bullet format;
- Some references are quite dated, especially those related to preservation technologies. Consider including more recent sources (2020–2025) to reflect the latest developments in biopreservation and clean-label strategies;
- Ensure consistency in reference formatting. Some entries seem to vary slightly in structure.
Author Response
Response to Reviewer Comment 1:
While the overall English is acceptable, some sentences are overly long or complex, which can hinder clarity. Consider breaking long sentences into two or rephrasing to improve readability.
Thank you for your valuable feedback regarding the language and clarity of the manuscript. We appreciated your observation about the overly long or complex sentences. Accordingly, we carefully revised the text to break up longer sentences and rephrased where needed to enhance readability and ensure better clarity throughout the manuscript.
Response to Reviewer Comment 2:
There are instances of redundancy or slightly awkward phrasing (e.g., "this approach aligns with the broader development of functional foods" could be simplified for impact).
Thank you for pointing this out. We have revised the sentence for clarity and conciseness.
Moreover, this approach supports the development of functional foods, products that offer health benefits beyond basic nutrition through bioactive compounds.
Response to Reviewer Comment 3:
The discussion section, though adequate, could benefit from a deeper comparison with previous findings in the literature. For example, highlight how the microbial activity and lipid oxidation trends observed in your study align or differ from related studies.
We thank the Reviewer for this valuable and constructive comment. We fully agree that a deeper comparison with relevant literature enhances the clarity and contextual strength of the discussion. In response, we have revised the text to explicitly integrate comparisons between our findings and those from recent studies, particularly focusing on the relationship between microbial inhibition and lipid oxidation.
A new paragraph has been added that discusses how the observed trends in microbial suppression and oxidative stability align with or differ from previously reported data. This includes references to studies by Papadochristopoulos et al. (2021), Yu et al. (2021), Domínguez et al. (2019), and Wu et al. (2022), which evaluate both microbiological and oxidative quality parameters in meat systems treated with natural biopreservatives. These comparisons support the interpretation of our findings and highlight the potential synergistic preservation effects of LAB and AAB supernatants.
We believe these additions significantly strengthen the discussion and better position our results within the context of current scientific knowledge.
Response to Reviewer Comment 4:
Consider addressing potential limitations of your experimental design, such as the use of only two bacterial mixtures or limited sample sizes (n=3 per group), and how this might affect generalizability.
We appreciate the Reviewer’s thoughtful observation regarding the scope of the experimental design. We would like to emphasize that the two bacterial mixtures selected for this study were the result of a series of preliminary investigations, during which various strain combinations and formulations were tested. These preliminary studies allowed us to identify the most promising mixtures in terms of antimicrobial and technological performance, which were then advanced to the current application-focused research.
To address the Reviewer’s comment and acknowledge this limitation, the following sentence has been added to the revised manuscript:
We believe that the research should be continued to expand the range of possible solutions, conditions, and include more food samples.
Response to Reviewer Comment 5:
The implications for industry application could be more explicitly addressed. How feasible is the use of these bacterial supernatants in commercial settings?
Thank you for this valuable suggestion. We have addressed this point by adding the following sentence to the conclusion section:
Given their demonstrated antimicrobial activity, ease of preparation, and cell-free nature, these supernatants offer a feasible and scalable approach for commercial application in meat processing environments.
This addition emphasizes the practical feasibility of implementing these supernatants in industrial settings.
Response to Reviewer Comment 6:
In some tables, superscripts indicating statistical differences are not well explained in the captions. Ensure every table includes clear footnotes for interpretation.
Thank you for noting this. We have revised the table captions to include clear footnotes explaining the superscripts used to indicate statistical differences. This ensures consistency and improves clarity for the reader.
Response to Reviewer Comment 7:
Although generally well-prepared, some tables are densely packed and could be made more reader-friendly with better spacing or simplified layouts.
Thank you for the suggestion. We agree that some tables are densely formatted; however, due to the volume of data and the need to present multiple variables side by side for comparison, simplifying the layout without omitting essential information proved challenging. We have, however, reviewed the formatting and made minor spacing adjustments where possible to improve readability while preserving the completeness of the data.
Response to Reviewer Comment 8:
Ensure all abbreviations used in tables and figures (e.g., TBARS, ORP, FPCA) are defined either in the table/figure legend.
Thank you for pointing this out. We have reviewed all tables and figures to ensure that abbreviations such as TBARS, ORP, and FPCA are clearly defined in the respective legends for clarity and ease of interpretation.
Response to Reviewer Comment 9:
The flow of the Results and Discussion sections could be improved by better separating findings from interpretations. Consider placing numerical data and findings first, then interpreting or comparing to literature in subsequent sentences.
Thank you for this valuable comment. We agree that improving the flow strengthens clarity. In revising the manuscript, we aimed to follow the structure suggested by the Reviewer—presenting results first, followed by interpretation and comparison with the literature. Although the Results and Discussion are integrated, we ensured this order is maintained within each subsection. Additional improvements were made in sections such as microbiological quality to better distinguish data from interpretation.
Response to Reviewer Comment 10:
The Conclusions could be more concise and focused on the main practical takeaways, perhaps in bullet format.
Thank you for your valuable feedback. In response to your suggestion, we have revised the Conclusions section to make it more concise and focused on the main practical takeaways. Where appropriate, bullet points were added to improve clarity and emphasize key outcomes. The updated Conclusions can be found in the revised manuscript.
Response to Reviewer Comment 11:
Some references are quite dated, especially those related to preservation technologies. Consider including more recent sources (2020–2025) to reflect the latest developments in biopreservation and clean-label strategies.
Thank you for your suggestion regarding the inclusion of more recent literature related to preservation technologies and clean-label strategies. However, we would like to clarify that the reference list already reflects a strong emphasis on recent research in the field.
Specifically, the manuscript includes:
- 3 references from 2025
- 12 from 2024
- 7 from 2023
- 12 from 2022
- 10 from 2021
- 4 from 2020
- 6 from 2019
In total, 54 out of the 71 references (over 75%) were published within the last six years (2019–2025), demonstrating a clear focus on the latest developments in biopreservation and clean-label strategies. The inclusion of earlier sources (from 2018 and prior) was intentional, as they represent foundational or widely cited works that remain relevant to the study’s scientific context.
We believe the current reference list provides a well-balanced and up-to-date foundation for the research presented.
Response to Reviewer Comment 12:
Ensure consistency in reference formatting. Some entries seem to vary slightly in structure.
Thank you for your observation. We have reviewed the reference list thoroughly and ensured consistency in formatting across all entries. This has been addressed in the current version of the manuscript.